# Postnatal Smoke Exposure Further Increases the Hepatic Nicotine Metabolism in Prenatally Smoke Exposed Male Offspring and Is Linked with Aberrant *Cyp2a5* Methylation

**DOI:** 10.3390/ijms22010164

**Published:** 2020-12-26

**Authors:** Khosbayar Lkhagvadorj, Zhijun Zeng, Karolin F. Meyer, Laura P. Verweij, Wierd Kooistra, Marjan Reinders-Luinge, Henk W. Dijkhuizen, Inge A. M. de Graaf, Torsten Plösch, Machteld N. Hylkema

**Affiliations:** 1University Medical Center Groningen, Department of Pathology and Medical Biology, University of Groningen, 9713 GZ Groningen, The Netherlands; k.lkhagvadorj@umcg.nl (K.L.); ZHJZENG88@outlook.com (Z.Z.); karolin.f.meyer@web.de (K.F.M.); lpverweij@gmail.com (L.P.V.); w.kooistra@umcg.nl (W.K.); m.luinge02@umcg.nl (M.R.-L.); 2GRIAC Research Institute, University of Groningen, 9713 AV Groningen, The Netherlands; 3Department of Pulmonology and Allergology, Mongolian National University of Medical Sciences, Ulaanbaatar 14210, Mongolia; 4Faculty of Science and Engineering, University of Groningen, 9713 GZ Groningen, The Netherlands; h.w.dijkhuizen@rug.nl; 5Department of Pharmacokinetics, Toxicology and Targeting, Groningen Research Institute of Pharmacy, University of Groningen, 9713 AV Groningen, The Netherlands; i.a.m.de.graaf@rug.nl; 6University Medical Center Groningen, Department of Obstetrics and Gynecology, University of Groningen, 9713 GZ Groningen, The Netherlands; t.plosch@umcg.nl

**Keywords:** epigenetics, CYP2A5, sex-difference, nicotine addiction, postnatal smoke exposure

## Abstract

Prenatal smoke exposure (PreSE) is a risk factor for nicotine dependence, which is further enhanced by postnatal smoke exposure (PostSE). One susceptibility gene to nicotine dependence is Cytochrome P450 (CYP) 2A6, an enzyme responsible for the conversion of nicotine to cotinine in the liver. Higher CYP2A6 activity is associated with nicotine dependence and could be regulated through DNA methylation. In this study we investigated whether PostSE further impaired PreSE-induced effects on nicotine metabolism, along with *Cyp2a5*, orthologue of *CYP2A6*, mRNA expression and DNA methylation. Using a mouse model where prenatally smoke-exposed adult offspring were exposed to cigarette smoke for 3 months, enzyme activity, mRNA levels, and promoter methylation of hepatic *Cyp2a5* were evaluated. We found that in male offspring, PostSE increased PreSE-induced cotinine levels and *Cyp2a5* mRNA expression. In addition, both PostSE and PreSE changed *Cyp2a5* DNA methylation in male groups. PreSE however decreased cotinine levels whereas it had no effect on *Cyp2a5* mRNA expression or methylation. These adverse outcomes of PreSE and PostSE were most prominent in males. When considered in the context of the human health aspects, the combined effect of prenatal and adolescent smoke exposure could lead to an accelerated risk for nicotine dependence later in life.

## 1. Introduction

Nicotine dependence is a complex disease, which is driven by an interplay of different causes and factors. Underlying mechanisms include genetic risk factors [1], as well as environmental exposures such as maternal smoking during pregnancy [2,3]. Human studies indicated that prenatal smoke exposure (PreSE) is associated with a higher probability of nicotine addiction [3], and a greater daily consumption of nicotine [4], as nicotine is the most prominent toxicant and psychoactive component in cigarette smoke. Cytochrome P450 (CYP) 2A6 is an enzyme responsible for the clearance of nicotine in the liver [5] by conversion of nicotine into cotinine [6], and *CYP2A6* mRNA expression has been shown to be increased after cigarette smoke exposure [7]. Higher CYP2A6 activity is associated with high smoking intensity [8,9], which is linked with alterations in the nicotine metabolism [10,11].

Among the members of the CYP2A family, mouse CYP2A5 is the orthologue to human CYP2A6 and is generally believed to be the major nicotine-to-cotinine metabolizer in the mouse, accounting for ~80% hepatic turnover [12]. Inhibition of CYP2A5 was shown to reduce the nicotine elimination rate along with prolonged nicotine-induced pharmacological effects [13], as nicotine circulated longer in the test body [14].

Thus far, aberrant DNA methylation patterns have been associated with deregulated gene levels, as shown in prenatally smoke-exposed children in birth cohort studies [15,16,17]. Prenatal programming is now thought to be one of the missing links to set the stage for metabolic disease development, albeit that the underlying mechanisms are not yet fully described. Previously, our studies have demonstrated that PreSE increased cotinine conversion rate, which was accompanied by aberrant methylation profiles and deregulated *Cyp2a5* mRNA expression in livers from fetal, neonatal and adult offspring [18]. In this study, we asked the question whether postnatal smoking (PostSE) would further enhance the effects of PreSE, as it has been shown that children of mothers who smoke during pregnancy will most likely be exposed to cigarette smoke while growing up [3,19] and have a higher chance to start smoking during adolescence [20].

## 2. Results

PreSE males had more cotinine formation compared to their controls (*p* = 0.03, Figure 1A), whereas in PreSE female offspring, cotinine levels were lower (*p* = 0.04, Figure 1B). Interestingly, in males, PostSE further enhanced the PreSE-induced cotinine levels, (positive interaction, (*p* = 0.002, Figure 1A), while no interactive effects were observed between PreSE and PostSE female offspring (Figure 1B). *Cyp2a5* mRNA levels in PostSE groups (*p* = 0.012, Figure 1C) were additionally increased, while no PostSE effect on mRNA was seen in female offspring (Figure 1D). Basal levels of cotinine formation and mRNA expression was higher in females than in males (*p* < 0.01, Figure 2A and *p* < 0.001, Figure 2B). A positive correlation was found between the cotinine conversion in all control groups and *Cyp2a5* mRNA levels (r = 0.67, *p* = 0.001; Table 1), while this correlation was not seen in both pre-and postnatal smoke exposed groups.

To investigate whether higher mRNA expression in PostSE male offspring groups was supported by epigenetic changes, *Cyp2a5* promoter methylation levels of six CpG sites (CpG-614, CpG-589, CpG-589, CpG-542, CpG-74, CpG-7, and CpG + 45) were assessed (Figure 3), as described in the Section 4. PreSE male offspring had higher methylation levels at CpG-589 (*p* = 0.02, Figure 3B) and CpG-74 (*p* = 0.01, Figure 3D) when compared to control male offspring, while no differences were observed in PreSE female offspring (Figure 3G–L). PostSE male offspring had higher methylation levels at CpG-589 (*p* = 0.01, Figure 3B), CpG-74 (*p* = 0.01, Figure 3D) and CpG-7 (*p* = 0.04, Figure 3E), whereas in female offspring, PostSE groups had higher methylation at CpG-542 (*p* = 0.01, Figure 3I) and CpG-7 (*p* = 0.02,Figure 3K). In contrast, PostSE males had lower methylated CpG-542 (*p* = 0.01, Figure 3C).

Basal levels of methylation of CpG-542, CpG-74 and CpG + 45 sites were higher in female offspring than in male offspring (all, *p*-values < 0.05, Figure 2C).

Based on these differentially methylated CpG sites in promoter region of *Cyp2a5*, we expanded our evaluation of mRNA expression and assessed whether DNA methyltransferase (DNMT) expression levels are affected by pre-and postnatal smoke exposures (Figure 4). It appeared that *Dnmt3a* mRNA levels were lower in the PostSE offspring, independent of their sexes (male: *p* = 0.01, Figure 4C, and female: *p* = 0.03, Figure 4D), while no differences were detected in PreSE offspring. Moreover, *Dnmt1* and *Dnmt3b* mRNA levels were not affected by PreSE and PostSE in both male and female offspring, as shown in Figure 4A,B,E,F.

In all PreSE groups, *Cyp2a5* mRNA levels correlated with CpG-614 methylation (r = 0.83, *p* = 0.003; Table 1), and negatively correlated with methylation at CpG-74 and CpG-7 (r = -82, *p* = 0.004 and r = -0.85, *p* = 0.004; Table 1). In PostSE males, *Cyp2a5* mRNA levels correlated with CpG + 45 methylation (r = 0.89, *p* = 0.00; Table 1), whereas in PostSE females, *Cyp2a5* mRNA levels correlated with CpG-614 and CpG-589 methylation (r = 0.76, *p* = 0.00; r = 0.93, *p* = 0.003, Table 1). *Cyp2a5* mRNA levels from the PreSE/PostSE male group correlated with CpG-589 methylation (r = 0.9, *p* = 0.02; Table 1). CpG-589 methylation levels correlated also with *Cyp2a5* mRNA levels in the PreSE females (r = 0.93, *p* = 0.00; Table 1).

## 3. Discussion

In our study, we found that PreSE increased cotinine formation levels, which was further enhanced by PostSE in PreSE offspring. This phenomenon was only observed in males. Higher nicotine metabolism was accompanied by higher *Cyp2a5* mRNA levels in PostSE male groups. In females, PostSE had no effect on the nicotine metabolism, and the PreSE group had lower cotinine formation than the other groups. Finally, *Cyp2a5* mRNA levels and differential DNA methylation patterns were more pronounced in PostSE male offspring than in PreSE offspring.

In this study, the increased nicotine metabolism in PreSE male offspring was further enhanced by personal smoke exposure of these mice. Extrapolating our observation to the human context, PreSE combined with PostSE could imply a serious risk for adverse smoking behavior, as a higher nicotine metabolism was associated with heavier smoking habits, and smokers who metabolize nicotine more rapidly may have difficulties quitting cigarette smoking than slower nicotine metabolizers [1,21,22]. Regarding the impact of PreSE alone on smoking behavior, maternal smoking during pregnancy selectively increased the probability that adolescent offspring would smoke and would persist in smoking. Concerning sex differences, an effect of PreSE on smoking behavior was stronger in adolescent daughters than in sons [3]. Other human studies support sex differences in nicotine dependence, showing that nonpharmacological effects, such as sex hormones and environmental factors have a stronger impact on drug and nicotine dependence in females than in males [23,24]. Considering that environmental factors are much more influential in humans as compared to laboratory animals, animal models are very useful to elucidate the biological relevance in nicotine dependence. Several experimental studies have assessed the impact of prenatal nicotine exposure on nicotine dependence in rat offspring [4,25]. Klein et al. described a preference of nicotine-containing water in prenatal nicotine exposed adolescent offspring, compared to non-exposed offspring. This effect was seen in males but not in female offspring [4]. In addition, prenatal nicotine-exposed hyperactive male offspring exhibited a significant increase in nicotine cholinergic receptors, while the effect was not seen in prenatal nicotine-exposed female offspring [25]. In accordance with our results, a potential combined effect of both pre- and postnatal smoke exposure on the nicotine metabolism is worth further exploration and may reveal important mechanisms of nicotine addiction in (early) adulthood.

In this study, we detected more PostSE-induced changes in *Cyp2a5* promoter methylation changes than PreSE-induced changes. In a human study, a similar result was described by Richmond and colleagues when comparing prenatally smoke-exposed newborns and adults [15]. In PreSE newborns, differentially methylated CpG sites were identified in *AHRR*, *MYO1G*, *GFI1*, *CNTNAP2*, *KLF13*, *CYP1A1* and *ATP9A* genes, which are known as smoke related genes. In that study, CpG sites in less than half of the smoking related genes have shown reversibility of methylation in PreSE offspring when compared to PreSE newborns. These findings could be an explanation for the current study showing that a majority of PreSE-induced CpG sites reverted to normal methylation “methylation recovery” after years of not being exposed to cigarette smoke [15,26,27], while differentially methylated CpG sites were more pronounced after PostSE. In addition, a current cigarette smoke exposure is known to be a powerful modifier for DNA methylation [28]. The effect of current smoke exposure has been associated with differential DNA in peripheral blood cells in multiple cross-sectional studies [29,30]. It should be noted in our study that PreSE-induced differentially methylated CpG sites of *Cyp2a5* have not been impaired by PostSE in both sexes.

In addition, we investigated the expression of DNMT enzymes in this mouse model. In mammals, DNA methylation is primarily catalyzed by three active DNMTs, which are maintenance methyltransferase (DNMT1) and *de novo* methyltransferase (DNMT3a and DNMT3b). Here, the expression of *Dnmt3a* mRNA was reduced by PostSE, while no significant changes were detected in PreSE offspring. The downregulatory effect of PostSE on expression of *Dnmt3a* is consistent with two previous in vitro studies, showing that exposure to cigarette smoke extract significantly reduced mRNA levels of *Dnmt3a* and *Dnmt3b* enzymes in embryonic orofacial cells [31] and mouse embryonic fibroblast cells [32]. The reduction in *Dnmt3a* and *Dnmt3b* mRNA levels were associated with decreased global and gene-specific DNA methylation patterns.

It is well known that smoke exposure effects on DNA methylation and gene expression are sex-dependent, as described by our research group [18,33,34] and others [28,29,30,35,36]. Similarly, we found that the distinct effects of PreSE and/ or PostSE on *Cyp2a5* methylation, mRNA levels and the nicotine conversion rate were different in male and female offspring. One interesting observation was that PreSE female offspring had a lower cotinine formation rate compared to controls. As a study in rats showed that early life smoke exposure caused impaired ovarian steroidogenesis [37], our result could be explained by a lower level of ovarian hormones in PreSE female offspring, leading to lower CYP2A5 activity. Another interesting observation was that female control offspring had a higher hepatic cotinine conversion and *Cyp2a5* mRNA expression as compared to males. This supports previous studies in humans [38,39,40] where female livers had a higher *CYP2A6* gene expression and enzyme activity than males [40]. This result is likely due to the effect of sex hormones on metabolic activity towards nicotine metabolism [40]. In addition to this, a study in rats demonstrated that ovariectomized female rats have a lower behavior motivation effect of nicotine intake than intact female rats [41,42].

Some limitations should be mentioned in the present study. Firstly, hepatic CYP2A5 protein levels were not measured to support a change in *Cyp2a5* mRNA levels. Secondly, RNA and DNA were isolated from whole liver tissue and a variation in the cellular composition of the liver in the different groups could be a confounding factor for the DNA methylation analysis. For the analyzed CpG sites, only the proximal CpG sites were investigated in *Cyp2a5* gene. However, although both the distal and proximal regions were described to regulate *Cyp2a5* in murine hepatocytes [43], the key activators of the *Cyp2a5* expression are within the proximal regions when compared to the distal region [44]. A final limitation is that blood and urine samples were not collected immediately after the final smoke exposure. They could have been an additional source for nicotine-to-cotinine conversion analyses as shown by in vivo mouse studies [12,14].

## 4. Materials and Methods

### 4.1. Animals & Smoke Exposure

A total of 48 female and 48 male C57BL/6J mice were obtained from Harlan (Horst, The Netherlands) at 6 weeks of age, housed under standard conditions with food and water provided ad libitum and with a 12-h light/dark cycle. The experimental setup was approved by the local committee on animal experimentation (DEC6589 B & C; University of Groningen, Groningen, The Netherlands) and under strict governmental and international guidelines on animal experimentation. Mainstream cigarette smoke was generated by using Teague10 (Tobacco and Health Research Institute of the University of Kentucky, Lexington, KY, USA). Over 7 days, randomly selected primiparous female mice were adjusted to cigarette smoke by increasing the number of smoked cigarettes step-wise (3R4 cigarettes; 2.45 mg nicotine/cigarette) from 2 to 5 every smoking session. At day 5 after the end of the second smoking session, all female mice were injected with pregnant mare’s serum gonadotropin (PMSG, 1.25 IU) to stimulate ovulation and at day 7 with human chorionic gonadotropin (hCG, 1.25 IU) to induce ovulation and housed on a 1:1 mating ratio with males overnight. Mating was confirmed by the presence of vaginal plug at the following morning.

Throughout gestation, female mice were housed in groups and exposed to two air or whole-body smoking sessions per day, 7 days per week. After delivery, dams were no longer exposed to cigarette smoke and housed individually for the following 8 weeks. Offspring (*n* = 46 from non-smoking mothers, *n* = 25 from smoking mothers) were whole-body exposed to air (16 males and 18 females) or smoke (19 males and 18 females) for 12 weeks, 5 days a week. A total of 71 adult offspring exhibited from 22 control (10 males and 12 females), 12 PreSE (6 males and 6 females), 24 PostSE (12 males and 12 females) and 13 Pre-and PostSE (7 males and 6 females), as shown in Figure 5. Offspring were euthanized at 20 weeks for collection of liver tissues. The material was immediately frozen in liquid nitrogen and stored at −80 °C until further use.

### 4.2. mRNA Expression and Pyrosequencing-Based Bisulfite PCR Analysis

DNA and RNA were isolated using the All Prep DNA/RNA Mini Kit (Qiagen, Cat No. 80204), according to the manufacturer’s protocol. Gene expression analysis in whole liver mRNA isolates was done via qPCR using qPCR MasterMix Plus (Eurogentec, Seraing, Belgium) with commercially available primers for *Cyp2a4/5* (Mm00487248_m1, TaqMan^®^ Gene Expression Assay, Applied Biosystems, Foster City, CA, USA), *Dnmt1*, *Dnmt3a* and *Dnmt3b* (Invitrogen, The Netherlands), and normalized to the housekeeping gene *Gapdh* (Mm99999915_m1, TaqMan^®^ Gene Expression Assay, Applied Biosystems, Foster City, CA, USA). Detection of amplification reactions was performed using the LightCycler^®^ 480 System (Roche Diagnostics GmbH, Mannheim, Germany), as previously described [18].

### 4.3. Isolation and Preparation of DNA of Liver Tissues for Bisulfite-Based Methylation Analysis

For the assessment of promoter methylation levels of *Cyp2a5*, bisulfite sequencing primers were designed using PyroMark assay design software (version 2.0, Qiagen). The selection of CpG-sites was based on manual identification of CpG-dinucleotides, using the ENSEMBL genome web browser (Ensembl 83: December 2015) and transcript location for the identification of gene promoter regions. Extracted genomic DNA from liver was converted with sodium bisulfite (EZ DNA methylation Direct™, Zymo Research, Irvine, CA), following the manufacturer’s instructions. In short, the bisulfite conversion was carried out in the dark at 98 °C for 10 min and 64 °C for 3.5 h, followed by desulphonation of the converted DNA. Gene amplification was done using HotStarTaq^®^ MasterMix (Kit Qiagen, Venlo, The Netherlands). CpG-sites were identified manually in the 600bp promoter region of the mouse Cyp2a5 gene (ENSMUSG00000005547), and assessment of DNA methylation levels was performed on the PyroMarkQ24 (Qiagen) instrument. Relative levels of methylation at each CpG-site were analyzed with PyroMark Q24 2.0.6 software. The used amplification and sequencing primers used in this study are listed in Table 2.

### 4.4. Microsome Preparation and In Vitro Assay for Nicotine Metabolism

Microsomes were obtained from homogenized livers of 20-week-old mice to measure nicotine metabolism, as previously described [5,18]. Briefly, the homogenized liver tissue was centrifuged at 9000× *g* for 20 min at 4 °C in phosphate buffer with EDTA. The supernatant fractions were collected and centrifuged at 100,000× *g* for 90 min at 4 °C. The resulting pellets (microsomes) were collected and re-suspended in phosphate buffer without EDTA. The collected supernatants (cytosolic fractions) were pooled for further use as a source of aldehyde oxidase activity for nicotine metabolism. The content of microsomal protein was determined in each sample with a reducing agent and detergent compatible (RC DC) protein assay (RC DC protein Assay, Bio-Rad, The Netherlands). In this study, we used the in vitro oxidation assay for nicotine-to-cotinine conversion, as adapted from previous mouse studies [14,45]. As cotinine is the major primary metabolite of nicotine metabolism in mice and humans [46], this approach has the clear advantage that the relative contribution of CYP2A5 (CYP2A6 in human) to cotinine formation can be estimated. The metabolism of nicotine follows two steps, the first of which is the conversion of nicotine to nicotine- Δ1′(5′)-iminium ion by CYP2A5 in the liver.

Thereafter, the nicotine- Δ1′(5′)-iminium ion was converted to cotinine by aldehyde oxidase, which is present in the cytosolic fraction. The liver microsomal protein was incubated with the pooled cytosolic fraction and nicotine containing solution. To ensure that the nicotine conversion was linear with time, and aldehyde oxidase availability was not rate-limiting, a 45-min incubation time was selected from the tested experimental durations (15, 30, 45 and 60 min), and 1mg/mL was selected as the optimal final concentration of cytosolic protein from a tested range between 0.25, 0.5, 1, 2, 3, 4, 6, and 8 mg/mL. These optimizations of nicotine-to-cotinine conversion were originally examined in the liver from PreSE embryonic and neonatal mice [18]. For the assay, a normalized 0.5 mg/mL liver microsomal protein and 80 µL cytosolic fraction with a protein content of 1.0 mg/mL were added to 80 µL of a freshly prepared nicotine containing solution, which is a mix of 4.4 µL S(-)-nicotine (≥99%, N3876, Sigma-Aldrich, St.Louis, MO, USA), 1.5 mL demineralized water and 28.5 mL of 100 mM sodium phosphate buffer pH 7.4. The mixture was supplemented with phosphate buffer (pH 7.4) to a total volume of 220 µL. After preheating this mixture at 37 °C for 5 min, 180 µL of freshly prepared NADPH in phosphate buffer containing 12.3 mM magnesium chloride (Sigma-Aldrich, St. Louis, MO, USA), 55.8 mM potassium chloride, 1.17 mM D-Glucose-6-phosphate (≥98%, G7250, Sigma-Aldrich, St.Louis, MO, USA) and 5.2 µL glucose-6-phosphate dehydrogenase (Roche, Germany; Grade II from yeast) were added to the mixture for supplying the energy to recycle the CYP’s active site after oxidation. This mixture (400 µL in total) was incubated at 37 °C for 45 min in an incubator for the actual biotransformation.

### 4.5. Nicotine Metabolite Analysis by HPLC and LCMS

For the analysis of the biotransformation products of nicotine metabolism, the samples were prepared as previously described by Messina et al. [6] and H. Raunio et al. [14]. The samples from the biotransformation experiment were alkalized by adding 20 µL of 10 M sodium hydroxide and mixing for 2–3 s on a vortex, to stop the reaction after the biotransformation assay, and to deprotonate the analytes. From this, the uncharged analytes were extracted in a two-step extraction. Each extraction step consisted of adding 1 mL of dichloromethane, mixing thoroughly for 60 s on a vortex, spinning for 3 min at 10,000 g to separate the layers, after which the organic (lower) phase was collected. An amount of 20 µL of 2.5 M hydrochloric acid was added to the combined dichloromethane fractions, as suggested by Massadeh et al. [47], before evaporating the solvent under a nitrogen atmosphere overnight at room temperature until dryness. Dried samples were reconstituted in 100 µL of HPLC buffer, which is a mix of 15% acetonitrile and 85% of 20 mM phosphate buffer, pH 3.0, containing 1.0 g/L heptane sulfonate sodium salt acid. The sample was analyzed by reverse phase HPLC with UV detection (245 nm), as shown in Figure 6A.

The separation was performed by HPLC-L, a Hitachi system consisting of an L2130 pump (set at isocratic elution of 0.6 mL/min) and an L2300 column oven set at 25 °C. To confirm the identity of the peak in the chromatogram a fragment of the cotinine fraction from HPLC system was sent to Interfaculty Mass Spectrometry Center (IMSC) at the University of Groningen. The sample was dried again under a nitrogen atmosphere and reconstituted in eluent (water and acetonitrile both containing formic acid) for the liquid chromatography separation. The sample was injected into the LCMS system, which was measured in positive mode from *m*/*z* 100–1000 and parallel reaction monitoring was included in the analysis for targeted analysis of the compounds with *m*/*z* 177. We identified that cotinine of the nicotine metabolism experiment displayed the same retention time as the cotinine standard. In the LCMS analysis, both the cotinine standard and the cotinine of the metabolism experiment were detected at an *m*/*z* of 177.10 (Figure 6B).

### 4.6. Statistical Methods

For statistical evaluation of the different groups, we chose two-tailed Mann-Whitney U-test for comparisons of two groups. For four groups, we performed a multiple linear regression analysis, the latter to distinguish between a positive interaction and a negative interaction. Spearman’s correlation test was used for correlation analysis. *p* values were adjusted by the Holm–Bonferroni method, and a value of *p* ≤ 0.05 was considered significant (GraphPad Prism 7.0 Software, San Diego, CA and SPSS Statistics 23, IBM, The Netherlands). Relative gene expression (2−ΔCt method) as well as mean percent methylation and standard error of the mean (SEM) were calculated in Microsoft^®^ Office Excel 2016.

## 5. Conclusions

This study presents that male offspring from mothers who were exposed to smoke during pregnancy have a higher nicotine metabolism which is further increased by 3 months of smoke exposure at adulthood. The higher nicotine metabolism is accompanied by higher *Cyp2a5* mRNA expression, which could be regulated with DNA methylation. Our study emphasizes that adverse outcomes of prenatal smoke exposure and postnatal smoke exposure are sex-dependent. When considering the impact of our observations on human health, the combined effect of prenatal and postnatal smoke exposure could lead to an accelerated risk for higher nicotine dependence later in life.

## Figures and Tables

**Figure 1 ijms-22-00164-f001:**
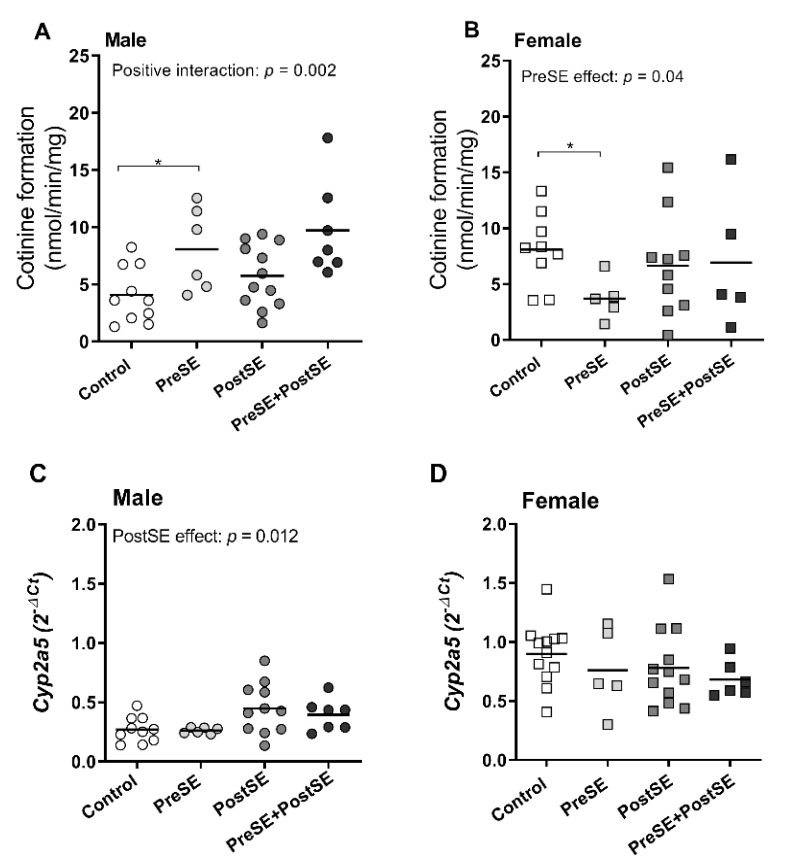
Cotinine formation rate of in vitro nicotine metabolism in the liver from (prenatal and postnatal) smoke exposed and control groups. Cotinine formation of hepatic nicotine metabolism and *Cyp2a5* mRNA expression were measured in liver from control, postnatal smoke exposed (PostSE), prenatally smoke exposed (PreSE), and the combined with prenatally smoke exposed and postnatal smoke exposed (PreSE/PostSE) mice. The cotinine formation rates (**A**, male and **B**, female) and mRNA expression (**C**, male and **D**, female) were measured by high performance liquid chromatography (HPLC) and qRT-PCR analysis as described in the Section 4. Circle (○) symbol(s) = male, square (□) symbol(s) = female. The linear regression analysis was performed among all groups. A positive interaction indicates a significant interaction between the effect of smoking during pregnancy and postnatal smoke exposure. Data are shown as individual values. If not stated otherwise, the comparison of shown groups was not significant. * *p* ≤ 0.05, (Mann–Whitney *U*-test). Open symbol(s) = control group, closed symbol(s) = (prenatally) smoke exposed group.

**Figure 2 ijms-22-00164-f002:**
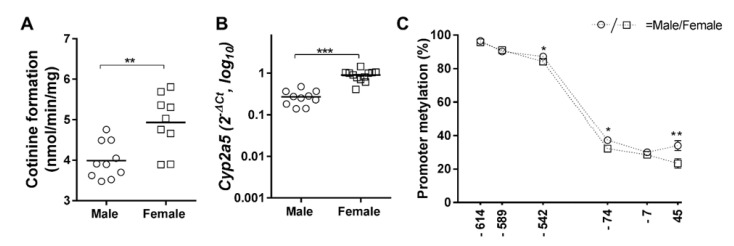
Comparisons of *Cyp2a5* mRNA expression and promoter methylation status between control male and female in the liver. (**A**,**B**) figures represent comparisons of cotinine levels and *Cyp2a5* mRNA expression between male and female in control liver and lung. (**C**) figure represents sex-dependent comparisons of *Cyp2a5* promoter regions between male and female in control liver and lung. Data of the 6 targeted CpG-sites are presented per sex and exposure as individual values with median as a horizontal line. Circle (○) symbol(s) = male, square (□) symbol(s) = female. If not stated otherwise, the comparison of shown groups was not significant. * *p* ≤ 0.05, ** *p* ≤ 0.01, *** *p* ≤ 0.001 (Mann–Whitney *U*-test). Open symbol(s) = control group.

**Figure 3 ijms-22-00164-f003:**
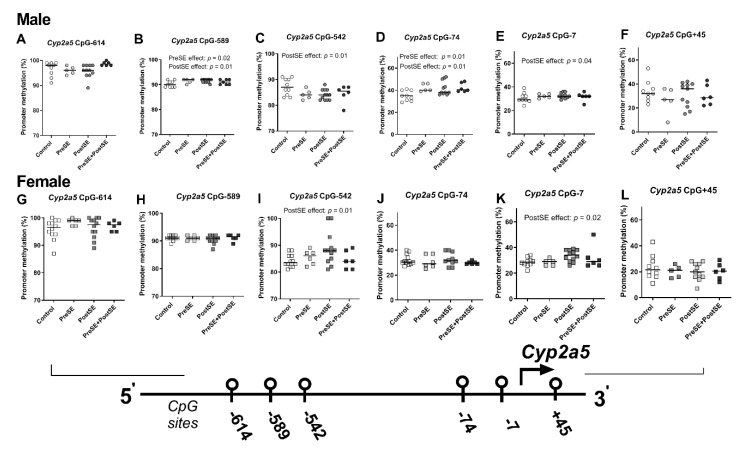
Sex-dependent Cyp2a5 promoter methylation in the liver from (prenatally) smoke exposed- and control groups. DNA was isolated from whole liver of male (**A**–**F**) and female (**G**–**L**) and from smoke exposed (prenatal and/or postnatal: closed symbols) and control groups (open symbols). DNA was subjected to bisulfite sequencing-based methylation analysis of the *Cyp2a5* promoter region and the percentage of DNA methylation was assessed. Data of the 6 targeted CpG-sites are presented per sex and exposure as individual values with median as a horizontal line. CpG-site annotations are relative to the ATG start codon. If not stated otherwise, the comparison of shown groups was not significant. A linear regression analysis was performed among allgroups. Circle (○) symbol(s) = male, square (□) symbol(s) = female. Open symbol(s) = control group, closed symbol(s) = (prenatally) smoke exposed group.

**Figure 4 ijms-22-00164-f004:**
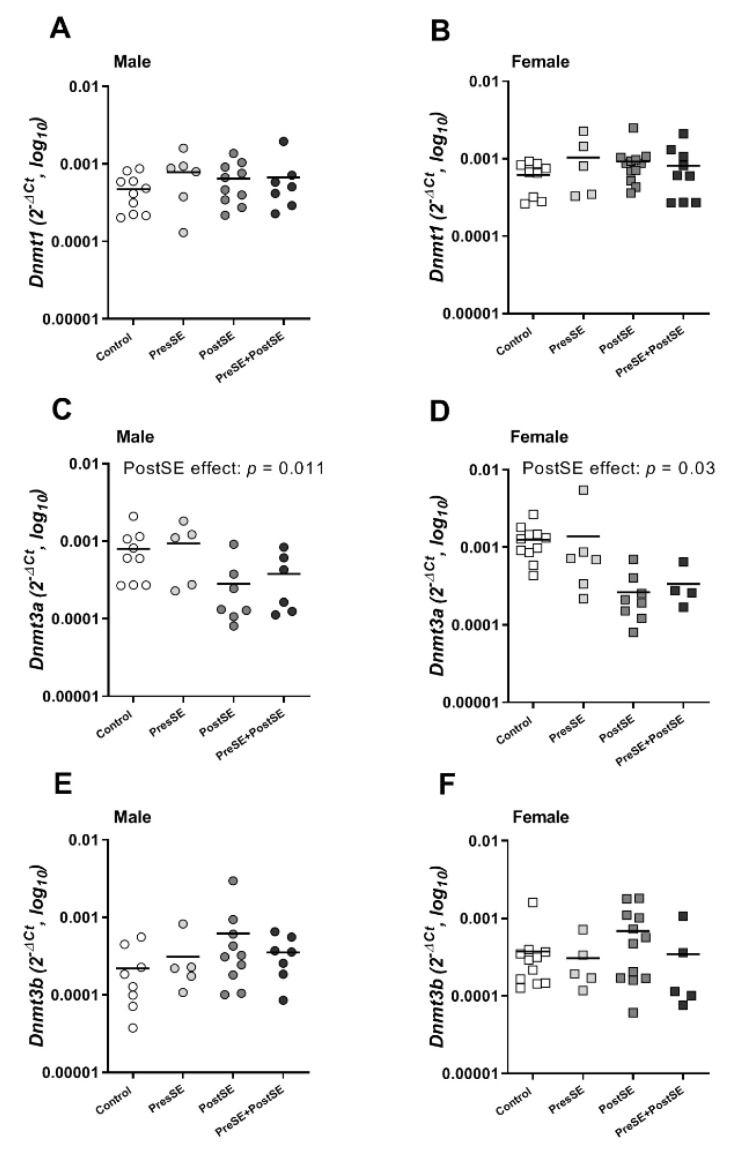
*Dnmt1*, *Dnmt3a* and *Dnmt3b* mRNA expressions were measured inwhole liver from control, PreSE, PostSE and PreSE/PostSE offspring. *Dnmt1* (**A**, male and **B**, female), *Dnmt3a* (**C**, male and **D**, female) and *Dnmt3b* (**E**, male and **F**, female) mRNA were quantified by qRT-PCR analysis as described in Section 4. A linear regression analysis was performed among all groups. Circle (○) symbol(s) = male, square (□) symbol(s) = female. If not stated otherwise, the comparison of shown groups was not significant.

**Figure 5 ijms-22-00164-f005:**
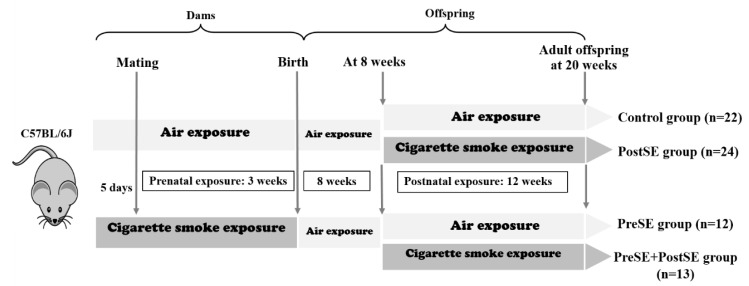
Study plan. The exposed offspring are divided into four group, which are air exposed offspring (control, *n* = 22), postnatal smoke exposed offspring (PostSE, *n* = 24), prenatally smoke exposed offspring (PreSE, *n* = 12), and the combined with prenatally smoke exposed and postnatal smoke exposed offspring (PreSE/PostSE, *n* = 13). Arrows indicate periods of exposure to air exposure (open box) and cigarette smoke exposure (closed box).

**Figure 6 ijms-22-00164-f006:**
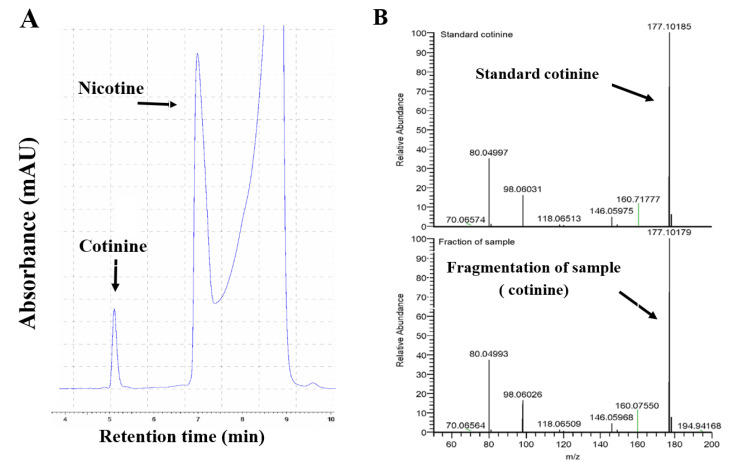
Cotinine formation rate of in vitro nicotine metabolism. The nicotine metabolism was measured by HPLC-L as described in the Section 4. (**A**) presents the HPLC chromatogram separation of the cotinine, which was produced in liver microsomes of a PreSE male mouse (**B**) presents the cotinine analysis by LCMS. Cotinine, corresponding with *m*/*z* 177 was detected both in a blank sample containing standard cotinine and in the microsome sample.

**Table 1 ijms-22-00164-t001:** Correlations between cotinine levels, *Cyp2a5* mRNA expression and promoter methylation in liver.

Correlation of/with			All (Male and Female)					Male					Female				
*Cyp2a5* (2^-dCT^)			All	All Control	All PreSE	All PostSE	All PreSE + PostSE	All	Control	PreSE	PostSE	PreSE + PostSE	All	Control	PreSE	PostSE	PreSE + PostSE
Cotinine formation (nmol/min/mg)		r	0.10	0.67	0.07	−0.03	−0.61	0.28	0.21	0.26	0.33	−0.25	−0.04	0.47	−0.40	−0.19	−0.63
		*p* value	ns	0.00	ns	ns	ns	ns	ns	ns	ns	ns	ns	ns	ns	ns	ns
*Cyp2a5* promotor methylation (%)	CpG−614	r	0.20	0.02	0.83	0.39	−0.31	0.06	0.38	0.20	−0.25	−0.06	0.44	0.12	0.82	0.76	0.53
		*p* value	ns	ns	0.00	ns	ns	ns	ns	ns	ns	ns	0.00	ns	ns	0.00	ns
	CpG−589	r	0.27	0.38	−0.10	0.23	0.48	0.31	0.03	−0.11	0.08	0.9	0.55	0.29	0.32	0.93	0.22
		*p* value	ns	ns	ns	ns	ns	ns	ns	ns	ns	0.02	0.00	ns	ns	0.00	ns
	CpG−542	r	−0.01	−0.36	0.22	0.12	0.11	−0.21	0.20	−0.60	−0.31	−0.29	0.04	0.04	−0.31	−0.20	0.68
		*p* value	ns	ns	ns	ns	ns	ns	ns	ns	ns	ns	ns	ns	ns	ns	ns
	CpG−74	r	−0.36	−0.17	−0.82	−0.26	−0.50	0.17	0.72	0.16	−0.16	−0.09	0.15	0.08	−0.60	0.26	0.34
		*p* value	0.00	ns	0.00	ns	ns	ns	ns	ns	ns	ns	ns	ns	ns	ns	ns
	CpG−7	r	−0.20	−0.33	−0.85	−0.03	−0.32	0.19	0.35	−0.67	−0.03	0.03	−0.35	−0.67	−0.63	−0.39	−0.46
		*p* value	ns	ns	0.00	ns	ns	ns	ns	ns	ns	ns	ns	ns	ns	ns	ns
	CpG + 45	r	−0.09	−0.01	−0.57	0.04	−0.55	0.52	0.25	−0.10	0.89	−0.26	0.11	0.12	0.40	0.10	−0.60
		*p* value	ns	ns	ns	ns	ns	0.00	ns	ns	0.00	ns	ns	ns	ns	ns	ns

*p*-values were adjusted by the Holm-Bonferroni method, ns= not significant.

**Table 2 ijms-22-00164-t002:** Sequences of primers used in bisulfite-based methylation analysis.

Gene	Targeted CpG-Sites	Sequence 5′-3′
*Cyp2a5*	CpG-614 to −542	For: TTTGTGTTTGTTTTGAGTGTTGGGATTA Rev: CCCCATCCACAACCATTCTT Seq: GTTGGGATTATAGGTTTATATTA Sequence to analyze: TTATATTYGATTTTTGGGAGTTTTTTAATGAAGAGGATTTTGAATTTAAGGATGYGAGAA GTGGAGATTT TAGGGTTATYGG
	CpG-74 to +45	For: AGTGGATAGTTTGGAGGTGAAAT Rev: ACAACTTTCCTAAAAACTTTCTCTACTTC Seq1: GGAGGTGAAATAGTTGTATAATTAA Seq2: TAGTTATTATTGTTTGTTTATTAT Sequence to analyze: S1: GATTAAAGTTYGTTTTTTTGTTTTTGGATGTATAAAAGTAAGTTAATT S2: TATYGTTATTATGTTGATTTTAGGATTTTTTTTGGTGGTTGTAGTGGTTTTTTTTAGYGTTTTGGTTTT

## Data Availability

Not applicable.

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
