# Peer review of "Postnatal Smoke Exposure Further Increases the Hepatic Nicotine Metabolism in Prenatally Smoke Exposed Male Offspring and Is Linked with Aberrant Cyp2a5 Methylation"

_ijms, 2020, doi:10.3390/ijms22010164_

Round 1
Reviewer 1 Report
The authors present a study using a murine mouse model where prenatally smoke-exposed miee were subsequently exposed to 12 weeks of smoke postnatally as adults. Outcones were nicrotine methabolis, Cyp2a5 gene expression and DNA methylation in the liver.
The major finding was that in male mice only, there was a synergistic effect between prenatal and postnatal smoke exposure with post natal exposure further increaing prenatal smoke expsoure induced cotinine levels and Cyp2a5 mRNA expression. The main conculsion ithe authors draw forom their resutls is that if extrapolated to humans, the combined effect of prenatal and adolescent smoke exposure could lead to an increased risk of nicotine dependence kn adulthood.
In general the paper is well written and the methods well described.
Major points
- Given that pre-conceptional smoking exposure (including of fathers) has been observed to lead to phenotypic effects in offspring in humans, do the authors feel that the pre-conceptional exposure of the female mice before mating may account for some of the sex specific differences observed in this study?
- One limitation of the study is that there was no wash-out period after the postnatal smoke exposure. Thus it is not possible to tell if the induction of gene expression and methylation by postnatal exposure (with or without pre-natal) is a transient effect due to direct smoke exposure or whether it leads to epigenetic programming of the liver with the differences persisting after cessation of smoke exposure. This should be acknowledged.
- Were cotinene levels measured in the male and female offspring to ensure the same level of exposure to cigarette smoke was occurring? I know the cages were exposed ot the same amount but are there differences such as respiration rate between males and females that may account for differences in response to nominally the same exposure?
Minor points:
1. Methods, in the model description section it is no exactly clear what the
Author Response
Reviewer 1.
The authors present a study using a murine mouse model where prenatally smoke-exposed mice were subsequently exposed to 12 weeks of smoke postnatally as adults. Outcomes were nicotine metabolic, Cyp2a5 gene expression and DNA methylation in the liver.
The major finding was that in male mice only, there was a synergistic effect between prenatal and postnatal smoke exposure with postnatal exposure further increasing prenatal smoke exposure induced cotinine levels and Cyp2a5 mRNA expression. The main conclusion it the authors draw from their results is that if extrapolated to humans, the combined effect of prenatal and adolescent smoke exposure could lead to an increased risk of nicotine dependence to adulthood.
In general, the paper is well written and the methods well described.
Major points
1C: Given that pre-conceptional smoking exposure (including of fathers) has been observed to lead to phenotypic effects in offspring in humans, do the authors feel that the pre-conceptional exposure of the female mice before mating may account for some of the sex-specific differences observed in this study?
1R: This is a very interesting question. We tend to agree that many risk factors for poor birth outcomes include smoking during the pre-conception period but we did not find studies to support this1,2. However, sex-dependent CYP2A5 enzyme activity is known to be related to female sex hormones. For example, in humans, pregnancy increases plasma nicotine and cotinine clearances3. A twin study with intravenous infusions of both nicotine and cotinine clearly shows that nicotine and cotinine clearances are higher in women than in men, as well as oral contraceptive use further accelerates nicotine and cotinine clearances in women4. Therefore, the female sex hormone may modulate nicotine-mediated behaviors reducing the relative impact of an adverse nicotine metabolism.
2C: One limitation of the study is that there was no wash-out period after the postnatal smoke exposure. Thus it is not possible to tell if the induction of gene expression and methylation by postnatal exposure (with or without pre-natal) is a transient effect due to direct smoke exposure or whether it leads to epigenetic programming of the liver with the differences persisting after cessation of smoke exposure. This should be acknowledged.
2R: The reviewer makes an interesting point on the persistence of epigenetic programming after cessation of smoke exposure in the liver. Previously, we reported the persistence of prenatally smoke-induced Cyp2a5 methylation, the Cyp2a5 mRNA expression as well as nicotine metabolism across three developmental stages (embryonic, neonatal and adulthood stages) in the same mouse model5. Indeed, prenatal smoke exposure induced increased methylation in almost all CpG sites of Cyp2a5 in fetal offspring, and this effect persisted in male neonatal and adult offspring. In addition to this, higher cotinine levels were detected in livers of male neonatal and adult offspring, as offspring have not been exposed after they were born. Generally, after smoking cessation, methylation levels gradually reversed to normal methylation between 3 and 6 months6. However, methylation of specific CpG sites was never foud to be restored after smoking cessation7. In the current study, we did no use the wash-out period after postnatal cigarette smoke exposure as we wanted to evaluate the postnatal (current) smoking effect in prenatally smoke-exposed offspring.
3C: Were cotinine levels measured in the male and female offspring to ensure the same level of exposure to cigarette smoke was occurring? I know the cages were exposed to the same amount but are there differences such as respiration rate between males and females that may account for differences in response to nominally the same exposure?
3R: In our study we were not able to measure the nicotine or cotinine levels in the offspring, as blood was drawn and mice were sacrificed 18h after the last cigarette smoke exposure. The lack of nicotine concentrations in blood supports a study from Zou et al. showing that nicotine has a half-life of approximately 10 minutes in C57Bl/6 mice8. The cotinine has a bit longer elimination half-life in plasma, which is approximately 38 minutes after 1 mg nicotine injection9. The urine samples were not collected in this study.
However, it seemed that male offspring had a stronger response to the smoke exposure than females, as smoke-exposed male offspring had a reduced body weight compared to the male control group (p=0.05), see Figure A. This effect was not seen in female offspring (Figure B). Housing conditions during cigarette smoke exposure were identical for all animals.
Bodyweight
- Pape K, Svanes C, Sejbæk CS, et al. Parental occupational exposure pre- and post-conception and development of asthma in offspring. Int J Epidemiol. 2020;(April):1-14. doi:10.1093/ije/dyaa085
- Brand JS, Gaillard R, West J, et al. Associations of maternal quitting, reducing, and continuing smoking during pregnancy with longitudinal fetal growth: Findings from Mendelian randomization and parental negative control studies. PLoS Med. 2019;16(11):1-24. doi:10.1371/journal.pmed.1002972
- Dempsey D, Jacob P, Benowitz N. Accelerated metabolism of nicotine and cotinine in pregnant smokers. J Pharmacol Exp Ther. 2002;301(2):594-598. doi:10.1124/jpet.301.2.594
- Benowitz NL, Lessov-Schlaggar CN, Swan GE, Jacob P. Female sex and oral contraceptive use accelerate nicotine metabolism. Clin Pharmacol Ther. 2006;79(5):480-488. doi:10.1016/j.clpt.2006.01.008
- Lkhagvadorj K, Meyer KF, Verweij LP, et al. Prenatal smoke exposure induces persistent Cyp2a5 methylation and increases nicotine metabolism in the liver of neonatal and adult male offspring. Epigenetics. 2020;00(00):1-16. doi:10.1080/15592294.2020.1782655
- Fragou D, Pakkidi E, Aschner M, Samanidou V, Kovatsi L. Smoking and DNA methylation: Correlation of methylation with smoking behavior and association with diseases and fetus development following prenatal exposure. Food Chem Toxicol. 2019;129(May):312-327. doi:10.1016/j.fct.2019.04.059
- Lee KWK, Richmond R, Hu P, et al. Prenatal exposure to maternal cigarette smoking and DNA methylation: Epigenome-wide association in a discovery sample of adolescents and replication in an independent cohort at birth through 17 years of age. Environ Health Perspect. 2015;123(2):193-199. doi:10.1289/ehp.1408614
- Petersen R, Norris J. A comparative study of the disposition of nicotine and its metabolites in three inbred strains of mice. Pharmacology. Published online 1984.
- Zhou X, Zhuo X, Xie F, et al. Role of CYP2A5 in the Clearance of Nicotine and Cotinine : Insights from Studies on a Cyp2a5 -null Mouse Model. J Pharmacol Exp Ther. 2010;Vol.3332:578-587. doi:10.1124/jpet.109.162610.

Reviewer 2 Report
The authors found that tobacco smoke exposure, prenatally, postnatally, or both, caused CYP2A5 methylation and hepatic microsomal nicotine metabolism. This is an interesting study. However, it seems that the authors did not put all data in the manuscript, or this is an unfinished study. Some concerns or issues in the present version lowered my enthusiasm to recommend for publishing.
- Were the urine and blood collected? How about cotinine levels in urine and blood? Is there a correlation between urine or blood cotinine and microsomal nicotine metabolism?
- How about CYP2A5 protein expression? While cyp2a5 mRNA was increased, CYP2A5 protein levels were also increased?
- Was the coumarin 7-hydroxylation (COH) activity in microsome measured? COH is considered as an activity marker of CYP2A5.
- What the significance of this study? Did the smoke exposure change the mouse behavior? Or lung function? Or liver metabolism function?
Minors:
Fig. 1 C and D, Y axis is in log scale, which make the differences in mRNA among groups unclear.
Author Response
Dear Editor, and reviewers,
We would like to thank the referees for their time and critical, but the valuable review of our manuscript “Postnatal smoke exposure further increases the hepatic nicotine metabolism in prenatally smoke exposed male offspring and is linked with aberrant Cyp2a5 methylation” (ijms-1027459).
We also would like to express our gratitude and recognition for the opportunity to submit a revised version.
In response to reviewer #2, we added a limitation in this study due to the lack of quantitative data on CYP2A5 protein levels (Lines 220-221). In addition, the Log10 scale of the Y-axis was removed in Figures 1C and D (Line 78).
We are optimistic that these narrative alterations now allow the acceptance of the manuscript by International Journal of Molecular Sciences
The concerns that were raised by the reviewers are addressed in a point-by-point fashion in attached PDF file.
Our response (R:) to reviewer comments (C:):

Round 2
Reviewer 2 Report
The answer to the reviewer's questions should be incorporated into the manuscript. If the authors want to put the answer to question 4 in another manuscript, the answers to the question 1 and 3 should be added.
Author Response
The concerns that were raised by reviewer#2 is addressed in a point-by-point fashion below.
Our response (R:) to reviewer comments (C:):
Reviewer 1C: The answer to the reviewer's questions should be incorporated into the manuscript. If the authors want to put the answer to question 4 in another manuscript, the answers to the question 1 and 3 should be added.
1R: We thank the reviewer for this comment.
Regarding the reviewer's question 1:
1C: Were the urine and blood collected? How about cotinine levels in urine and blood? Is there a correlation between urine or blood cotinine and microsomal nicotine metabolism?
1R: We have added a sentence in the discussion part that “A final limitation is that blood and urine samples were not collected immediately after the final smoke exposure. They could have been an additional source for nicotine-to-cotinine conversion analyses as shown by in vivo mouse studies[4][5]”
(Lines 222-224).
Regarding the reviewer's question 2:
2C: How about CYP2A5 protein expression? While cyp2a5 mRNA was increased, CYP2A5 protein levels were also increased?
2R: We have added a sentence in the discussion part where the limitations of the study are discussed. In lines 215-216 we write: “ Firstly, hepatic CYP2A5 protein levels were not measured to support a change in Cyp2a5 mRNA levels”.
Regarding the reviewer's question 3:
3C: Was the coumarin 7-hydroxylation (COH) activity in microsome measured? COH is considered as an activity marker of CYP2A5.
3R: We agree that CYP2A5 (CYP2A6 in human) activity can be measured using coumarin as the main metabolite. However, in 1954, coumarin was banned by the US Food and Drug Administration (FDA) in food additives and cosmetics as a result of studies indicating that coumarin has toxic effects on the liver of animals[6]. In addition, the use of coumarin in nicotine metabolism assays was not favorable in mouse studies, as CYP2A5 works efficiently in-oxidation assays for nicotine-to-cotinine conversion in C57BL/6 mice[1].
Based on the reviewer’s comment 3, we added the following sentences in the method section (Lines 288-293).
“In this study, we used the in vitro-oxidation assay for nicotine-to cotinine conversion, adapted from previous mouse studies[2], [7]. As cotinine is the primary metabolite of nicotine metabolism in mice and humans[3], this approach has the clear advantage that the relative contribution of CYP2A5 (CYP2A6 in human) to cotinine formation can be estimated. The metabolism of nicotine follows two steps, the first of which is the conversion of nicotine to nicotine- Δ1'(5')-iminium ion by CYP2A5 in the liver”.
[1] E. C. K. Siu and R. F. Tyndale, “Characterization and Comparison of Nicotine and Cotinine Metabolism in Vitro and in Vivo in DBA/2 and C57BL/6 Mice,” Mol. Pharmacol., vol. 71, no. 3, pp. 826–834, 2006.
[2] H. Raunio et al., “Nicotine metabolism and urinary elimination in mouse: In vitro and in vivo,” Xenobiotica, vol. 38, no. 1, pp. 34–47, 2008.
[3] D. Dempsey et al., “Nicotine metabolite ratio as an index of cytochrome P450 2A6 metabolic activity,” Clin. Pharmacol. Ther., vol. 76, no. 1, pp. 64–72, 2004.
[4] X. Zhou et al., “Role of CYP2A5 in the Clearance of Nicotine and Cotinine : Insights from Studies on a Cyp2a5 -null Mouse Model,” J. Pharmacol. Exp. Ther., vol. Vol.3332, pp. 578–587, 2010.
[5] H. Raunio et al., “Nicotine metabolism and urinary elimination in mouse: In vitro and in vivo,” Xenobiotica, vol. 38, no. 1, pp. 34–47, 2008.
[6] R. T. J. and H. K. M. L. W. Hazleton, T. W. Tusing, B. R. Zeitlin, “Toxicity of coumarin,” J. Pharmacol. Exp. Ther.
[7] E. C. K. Siu, D. B. Wildenauer, and R. F. Tyndale, “Nicotine self-administration in mice is associated with rates of nicotine inactivation by CYP2A5,” Psychopharmacology (Berl)., vol. 184, no. 3–4, pp. 401–408, 2006.
